# MiDRM*pol*: A High-Throughput Multiplexed Amplicon Sequencing Workflow to Quantify HIV-1 Drug Resistance Mutations against Protease, Reverse Transcriptase, and Integrase Inhibitors

**DOI:** 10.3390/v11090806

**Published:** 2019-08-30

**Authors:** Shambhu G. Aralaguppe, Anoop T. Ambikan, Manickam Ashokkumar, Milner M. Kumar, Luke Elizabeth Hanna, Wondwossen Amogne, Anders Sönnerborg, Ujjwal Neogi

**Affiliations:** 1Division of Clinical Microbiology, Department of Laboratory Medicine, Karolinska Institutet, Huddinge, 14186 Stockholm, Sweden; 2Department of HIV/AIDS, National Institute for Research in Tuberculosis, Indian Council of Medical Research, Chennai 600031, India; 3Blackbuck Technologies, Porur, Chennai 600116, India; 4Department of Internal Medicine, School of Medicine, Addis Ababa University, 2380 Addis Ababa, Ethiopia; 5Division of Infectious Diseases, Department of Medicine Huddinge, Karolinska Institutet, 14186 Stockholm, Sweden

**Keywords:** MiDRM*pol*, high-throughput sequencing, drug resistance mutation, antiretroviral therapy, surveillance

## Abstract

The detection of drug resistance mutations (DRMs) in minor viral populations is of potential clinical importance. However, sophisticated computational infrastructure and competence for analysis of high-throughput sequencing (HTS) data lack at most diagnostic laboratories. Thus, we have proposed a new pipeline, MiDRM*pol*, to quantify DRM from the HIV-1 *pol* region. The gag-vpu region of 87 plasma samples from HIV-infected individuals from three cohorts was amplified and sequenced by Illumina HiSeq2500. The sequence reads were adapter-trimmed, followed by analysis using in-house scripts. Samples from Swedish and Ethiopian cohorts were also sequenced by Sanger sequencing. The pipeline was validated against the online tool PASeq (Polymorphism Analysis by Sequencing). Based on an error rate of <1%, a value of >1% was set as reliable to consider a minor variant. Both pipelines detected the mutations in the dominant viral populations, while discrepancies were observed in minor viral populations. In five HIV-1 subtype C samples, minor mutations were detected at the <5% level by MiDRM*pol* but not by PASeq. MiDRM*pol* is a computationally as well as labor efficient bioinformatics pipeline for the detection of DRM from HTS data. It identifies minor viral populations (<20%) of DRMs. Our method can be incorporated into large-scale surveillance of HIV-1 DRM.

## 1. Introduction

Genotypic resistance testing (GRT) to monitor antiretroviral therapy (ART) of human immunodeficiency virus type 1 (HIV-1) infection is challenging due to drug resistance mutations (DRMs) in minor viral populations not detected by Sanger sequencing, although the impact of such DRMs on clinical outcome is still under debate [1]. A major challenge for the implementation of high-throughput sequencing (HTS) in clinical practice is the need of bioinformatics expertise, since there is still no consensus on the standardization of an efficient and easy pipeline for data analysis. However, advances in HTS technology have facilitated sufficient throughput data and sensitivity to detect very rare viral DRMs [2]. On the other hand, HTS-based GRT has a few limitations that need to be addressed, including short and error-prone reads [2], low-abundance interpretation of variant data, and a lack of standardization of data analysis protocols/pipelines [3]. 

In this study, we evaluated a simplified bioinformatics pipeline, named MiDRM*pol*, for analysis of HTS data which is cost, labor, and computationally efficient. The Fastq files generated by HTS can be used as input to get easily inferable diagnostic reports and can be applied in large-scale surveillance. It can be used without prior knowledge of HTS or bioinformatics. Here, we restricted the analysis of DRMs to the *pol* region of the HIV-1 genome, including the targets of the vast majority of presently approved HIV-1 drugs: the reverse transcriptase, protease, and integrase.

## 2. Materials and Methods 

### 2.1. Clinical Specimens, Plasmids, and Control Virus

Plasma samples were obtained from three cohorts originating from Sweden (*n* = 60), India (*n* = 10), and Ethiopia (*n* = 17). All patients were treatment naïve except 10 Swedish subjects, of whom only 2 were given ART at the time of plasma sampling. The majority (30/50; 60%) of the Swedish treatment-naïve subjects had been infected outside the country. Eight of the 10 Indian samples were from children who were infected through mother-to-child transmission (MTCT). Besides, to deduce the reliability of the method, a standard molecular clone (pMJ4) [4] was used as described in [5]. 

### 2.2. High-Throughput Sequencing

The cDNA was prepared using gene-specific primers and the HTS was carried out as described by us [5]. The F1 fragment (*Gag-to-vpu)* (HXB2, 790-6231) was amplified and gel-purified using the QIAamp gel extraction kit (Qiagen, Venlo, The Netherlands). Briefly, the purified amplicon was fragmented on the Covaris S200 at 300 bp for 75 s with peak power-50 and cycle/burst-200, and the library was prepared using NEBNext^®^ Ultra™ DNA Library Prep Kit for Illumina^®^ (New England Biolabs, Ipswich, MA, USA) with multiplexed NEB next adaptors. The samples were then pooled together in sets of 48 samples along with other unrelated nonviral indexed libraries. Paired-end sequences of read length 250 bp were carried out on the Illumina HiSeq2500. 

### 2.3. Bioinformatics Pipeline for Analysis

The raw reads were adapter-trimmed using the Cutadapt v1.8 program using the default setting (error rate of <0.1) [6], followed by removal of the low-quality bases (Phred value score of <Q30) by Sickle version 1.33 [7]. Duplicate reads were removed using FastUniq [8]. The processed reads were aligned against the gene nucleotide sequence of POL from HIV. This alignment was performed in local mode using Bowtie2 [9] to identify reads originating from the POL gene. Then, these reads were further aligned with the POL protein sequence using the blastx program from Blast package [10]. For each read, the best blast hit was chosen and all amino acid substitutions were counted. Though there were multiple amino acid substitutions at each position in the POL protein sequence, only the most-frequent substitution was considered, and the fraction of the most-frequent amino acid substitution was estimated. The complete script is available at GitHub at https://github.com/neogilab/MiDRMPol.

### 2.4. Standard Population Sequencing

The *ViroSeq* HIV-1 genotyping system had been used for the Swedish patients as a part of routine clinical practice. HIV-1 GRT by population sequencing (GRT-PS) was also performed for the Ethiopian patients using an in-house method [11] but not for Indian patients due to insufficient sample availability. 

### 2.5. HIV-1 Subtyping and Phylogenetic Analysis

Consensus *pol* sequences were generated from the base count data with the occurrence of a particular nucleotide of >50% in a given position and subtyped by REGAv3 [12] and COMET-HIV [13], followed by phylogenetic analysis using MEGA7.

### 2.6. Drug Resistance Mutations

For treatment-naïve individuals, DRMs was described using definitions for surveillance of transmitted HIV-1 drug resistance, 2009 Update (sDRM2009) [14]. Mutation lists provided in the Stanford University HIV Drug Resistance Database [15], accessed on 6 January 2018 [15,16], was used for describing major integrase mutations, including Q148H, E138K, T66I, and Q148H. The DRMs of treatment-experienced patients were interpreted based on the Stanford University HIV Drug Resistance Database [15].

### 2.7. Availability of Data

The GenBank IDs for the sequences are available with the following accession numbers: for the Indian cohort: KX069219-228; for the Swedish cohort: MF373124-206; and for the Ethiopian cohort: KP411823-826, 828 and KP411830-845. Fastq files: Submission ID: SUB5359871, BioProject ID: PRJNA529776.

### 2.8. Ethics Approval and Consent of Participants

Ethical permissions were obtained from the respective sites. Swedish samples: Regional Ethics Committee Stockholm (Dnr: 2006/1367-31/4); Ethiopian samples: the Ethiopian Science and Technology Agency (Ref. No. RPHE/126-83/08), and the Drug Administration and Control Authority of Ethiopia (Ref. No. 02/6/22/17); Indian samples: National Institute for Research in Tuberculosis Institutional Ethics Committee (NIRT IEC No. 2009009). The patients’ information was anonymized and delinked before analysis. 

## 3. Results

### 3.1. Coverage and Error Rate

The amino acid coverage at each postion of the HIV-pol region was calculated in terms of quartiles. The range of first quartile was 3836x–12,725x whereas for third quartile, it was 15,403×–36,451x. All the DRM positions had median coverage >5000x (Figure 1a). To estimate the intrinsic error introduced by the laboratory workflow, we sequenced pMJ4, where the error rate was found to be <1% at the nucleotide level [5]. In addition, we determined the median error rate at the amino acid level, which was 0.2% (0–1.8%) (Figure 1b).

### 3.2. Subtyping and Phylogenetic Analysis

Based on the phylogenetic analysis, we found that the majority of the viral isolates (*n* = 78, 90%) were subtype C, of which 10 isolates were from treatment-experienced individuals. Seven (8%) isolates were subtype B. In addition, we identified one HIV-1A1 and one 01_AE. No cross contamination was found (Figure 2). Two Swedish samples (SE70 and SE73) clustered together and were epidemiologically linked.

### 3.3. Identification of DRM by GRT-HTS

DRMs were identified and quantified in 23/87 patients (Table 1). In the Swedish cohort, 9/50 (18%) therapy-naïve patients had DRMs (M46I: *n* = 3) (M184I, T215S, or K219R: *n* = 3); major--integrase strand transfer inhibitors (INSTI) mutation Q148H or E138K: *n* = 2)). Among Indian patients, 5/10 (50%) had reverse transcriptase inhibitors (RTI) DRM, and all 5 were MTCT patients. No DRMs were detected in horizontally infected patients. Among Ethiopian patients, 6/17 (35%) had DRMs, of whom 5 had a single-class DRM and one had both nucleoside reverse transcriptase inhibitors (NRTI) (M184I) and INSTI (Q148H) DRM.

Among the 10 treatment-experienced patients from the Swedish cohort, only two patients were on ART at the time of plasma sampling, although they had high viral RNA levels, indicating that they did not take the drugs and no DRMs were found. In three patients, DRMs were found. One patient (SE03; HIV-1C) who had terminated nevirapine, tenofovir, and raltegravir therapy one month earlier had D67G and L74I. This patient had earlier been given abacavir-containing therapy. Patients SE70 and SE73 (HIV-1C) had stopped cART ABC + DDI + EFV and d4T + 3TC + NVP, respectively, a few months before plasma sampling. These subjects were epidemiologically linked and were identified with the same set of RTI mutations: D67N, L210W, T215Y, K103N, and G190A.

### 3.4. Comparison of DRM Detected by GRT-PS and GRT-HTS

While comparing GRT-PS and GRT-HTS among the Swedish and Ethiopian patients, additional DRMs were detected by GRT-HTS (Table 1). Among the treatment-naïve individuals (*n* = 77), GRT-HTS detected DRMs in 26% (20/77) of the samples (e.g., M184I (*n* = 4), T125S (*n* = 1), and M230L (*n* = 1)), whereas GRT-PS detected them in only one naïve patient (M230L). 

In treatment-experienced patients (*n* = 10), DRMs were detected in two and three samples by GRT-PS and GRT-HTS, respectively. In one sample (SE73), Y188YC was detected by GRT-PS and not by GRT-HTS. One of the Indian MTCT samples (IN07) had several major NRTI and NNRTI DRMs at frequencies of <20%. GRT-HTS also identified major INSTI mutations at low frequencies (e.g., Q148H, E138K, and T66I), which were undetectable by GRT-PS. In two samples (SE73 and SE31), GRT-PS did not detect DRMs which were >20%. 

When comparing MiDRM*pol* and PASeq outputs for 23 samples, the same DRMs with comparable frequencies were reported for 13 sequences (Table 1). In six samples (SE03, SE07, ET115, ET122, ET159, and ET171), only MiDRM*pol* reported minor DRMs (frequency: 1.0%–4.8%). In four samples, PASeq did not deliver results due to the large file size (the file upload limit of PASeq is 1 GB). 

In detail, the discrepancy was due to M184I (1.0%–1.5%), which is considered to be a DRM in MiDRM*pol* but not in PASeq, in three samples (ET115, ET159, and ET171). In sample SE03, D67G and L74I (2.0% and 1.9%, respectively); in samples SE07 and ET171, Q148H (1.1% and 1.2%, respectively); and in sample ET122, M230L (4.8%) were reported by MiDRM*pol* but not by PASeq. This is possibly because of the FastUniq tool used in MiDRM*pol*, which removes duplicate reads, due to which the minor populations of mutations are enriched. 

In sample 24C, T215A (99.7%) and T69N (41.7%) were not detected in our pipeline because we do not consider T215A and T69N as resistance mutations against any of the RTIs. One of the mutations (D67E) was detected at 77.1% in PASeq but not detected in our pipeline.

## 4. Discussion

In this study, our in-house pipeline, MiDRM*pol*, was evaluated for the identification of DRMs in sequences obtained by HTS. Patients from Swedish, Indian, and Ethiopian cohorts, representing diverse subtypes, were included. The MiDRM*pol* pipeline was successful in detecting all major and minor DRMs. Identification of mutations such as M184I, which appears at the very early stage of the evolution of M184V, and T215S, a revertant mutation, by GRT-HTS shows that it can detect the early evolution of DRM and earlier exposure of cART, which can provide further information about, for example, pretreatment drug resistance (PDR) in global surveillances of the spread of resistant HIV-1. Although GRT-HTS detects HIV DRMs more sensitively than GRT-PS, the generalized application of such methods for clinical DR monitoring is not currently feasible due to the high cost per sample. However, the cost per megabase of raw data of DNA sequences has decreased 370-fold, from US$5.200 in 2001, to US$0.014 in 2015 [17]. Equipment and maintenance costs, however, remain unaffordable for many low-income countries. The most important limitation, however, is the lack of automated, validated, and robust but simplified bioinformatic analyses coupled with HIV-1 resistance interpretations to enable NGS use and interpretation by laboratory technicians, but even this is improving rapidly [18]. Currently, for our approach, the cost per sample was US$130–US$150 when 24 samples were pooled in a single run [5]. This can be further reduced by multiplexing a higher number of samples. MiDRM*pol* would cut the costs down by excluding the necessity of bioinformaticians to analyze HTS data. 

DRMs were identified 8 times more often by GRT-HTS than by GRT-PS, which is in agreement with another study, which observed that 30%–50% of transmitted drug resistance (TDR) mutations were not identified by GRT-PS [19,20]. An investigation of the established pipelines PASeq and HyDRA showed minimal advantage when using HTS data over Sanger sequencing in identifying low-frequency DRMs (≥5%) in virological failure individuals treated with integrase inhibitors [21]. In our earlier study of an Ethiopian cohort, we identified an increased risk of treatment failure in patients who had pretherapy minor RTI mutations [22]. To the best of our knowledge, no bioinformatics pipeline interprets the success of ART. It is important that the outcome of the result generated through the pipeline is used to predict the treatment outcomes and guide patient management in clinical practice [23]. The impact of drug-resistant variants does not depend only on the frequency; rather, the mutational load (Equation (1)) determines more precisely the absolute burden of the resistant variants [24,25]. Thus, at a high viral load, even <1% of minor NNRTI DRM variants have been claimed to have the potential to overcome the drug pressure [26,27].
(1)Mutationial load = HIV RNA copies per mL ×Mutant variant frequency.

However, the pipeline MiDRM*pol* was designed to identify >1% of minor DRM populations, independent of the subtype. Although 10 of the patients from Sweden were treatment experienced, only 2 of them claimed to be undergoing treatment but had most likely not taken the drugs. Only 3 of the 10 patients had detectable DRMs despite earlier treatment failures. It should, however, be noted that ART had been terminated just a few months before sampling, suggesting that the DRMs had not yet had time to disappear. In other patients, the treatment had been terminated several months or years before. Thus, although no precise analysis can be done due to lack of samples during earlier treatment failure, our data indicate that even the use of a very sensitive HTS may fail to detect DRMs which had developed at earlier treatment failures. 

In the Indian cohort, we identified minor DRMs in 2 out of 10 treatment-naïve individuals who were vertically infected. Eight minor RTI DRMs, including K65R (3.0%), M184V (7.5%), and V106M (3.9%), were detected in one infant (IN07), as a result of the transmission of the resistant virus from the mother. Thus, our data show that GRT-HTS improves detection of vertically transmitted DRMs in infants [28]. 

Similar algorithms for determining the frequency of DRM variants are available, for example, PASeq [29], a web service which uses various tools, and CoVaMa (Co-Variation Mapper), which is scripted in Python and takes HTS alignment data and populates large matrices of contingency tables [30]. Other available automated pipelines for analysis of sequences obtained through HTS have been summarized in a comparative study [31]. We compared our pipeline with PASeq and observed concordance at the DRM level, although our pipeline also detected M184I mutations in treatment-naïve individuals at minor populations. 

## 5. Conclusions

In this study, we developed an efficient and simplified high-throughput amplicon sequence analysis pipeline named MiDRM*pol*, which does not require onsite bioinformatics expertise. This pipeline can detect minor DRMs in viral populations at a >1% cutoff and was successfully evaluated in three cohorts consisting of various subtypes. The reliability was analyzed by comparing the sequences generated through both GRT-HTS and GRT-PS. In all of our cohorts, MiDRM*pol* identified DRMs against the three drug classes in minor viral quasi-species at analysis of sequences obtained through HTS. The complete script is available at GitHub at https://github.com/neogilab/MiDRMPol. Our simplified online pipeline for HTS data of the HIV-1 *pol* gene would be a valuable tool in the use of diagnostics of drug resistance variants before initiation of ART and after ART failure in optimizing subsequent regimens. 

## Figures and Tables

**Figure 1 viruses-11-00806-f001:**
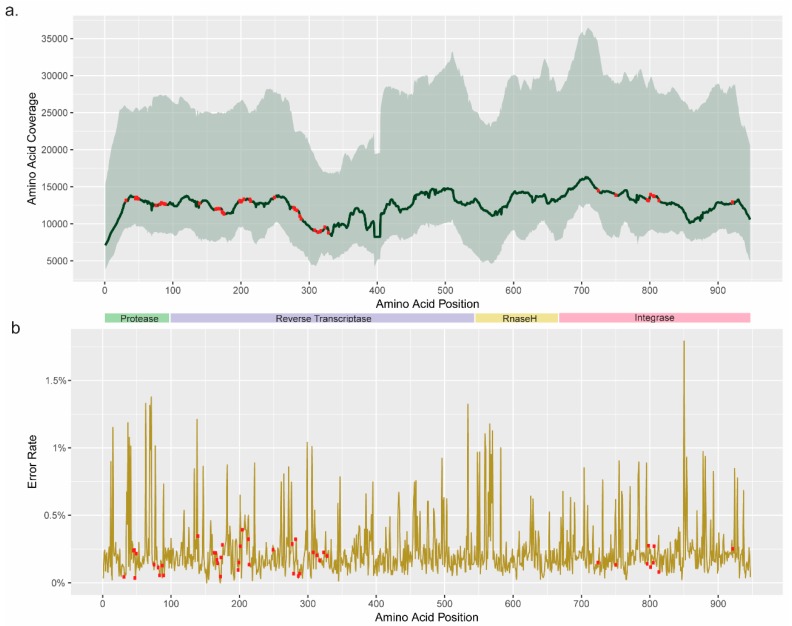
Coverage plot and error calculation: (**a**) The median coverage in each of the positions is indicated by the a dark green line. The light-green shade is the range of the coverage. (**b**) The error was calculated from the amino acid sequences of pMJ4 (NCBI# AF321523). Red dots indicate the drug resistance mutation (DRM) positions.

**Figure 2 viruses-11-00806-f002:**
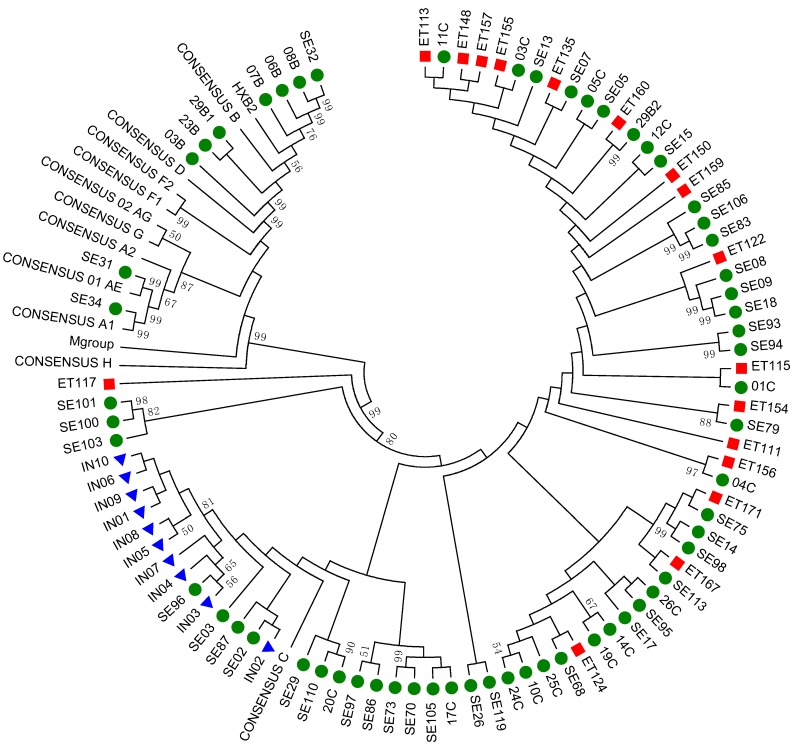
The phylogenetic analysis of the gag-pol sequences. A total of 100 sequences were used in the analysis (87 samples and 13 consensus sequences of gag-vpu from diverse HIV-1 subtypes downloaded from HIVdb). Green circles represent the Swedish cohort (*n* = 60), blue triangles represent the Indian cohort (*n* = 10), and red squares represent the Ethiopian cohort (*n* = 17). The reference sequences of gag-pol from various genetic subtypes—A1, A2, B, C, D, F1, F2, G, H, AE, and AG—were downloaded from the Los Alamos HIV database. After gap-stripping, there were a total of 3096 nucleic acid positions in the final dataset. The bootstrap consensus tree inferred from 1000 replicates is taken to represent the evolutionary history of the taxa analyzed. The tree was drawn to scale, with branch lengths in the same units as those of the evolutionary distances used to infer the phylogenetic tree. The evolutionary distances were computed using the maximum composite likelihood method and are in the units of the number of base substitutions per site.

**Table 1 viruses-11-00806-t001:** Comparison of DRMs identified by high-throughput sequencing or population sequencing as well as in PASeq and MiDRM*pol* outputs.

Cohort	PID	Treatment	MiDRM*pol*				PASeq				GRT-PS	
			PI	NRTI	NNRTI	INI	PI	NRTI	NNRTI	INI	NRTI	NNRTI
**Swedish**	**SE03**	Experienced	None	**D67G (2.0%)**	None	None	None	None	**K103R (16.9%)**	None	None	None
				**L74I (1.9%)**								
	**SE70**	Experienced	None	**D67N (99.7%)**	**K103N (99.8%)**	None	None	**D67N (99.9%)**	**K103N (99.9%)**	None	**D67N**	**K103N**
				**L210W (99.5%)**	**G190A (99.6%)**			**L210W (99.8%)**	**G190A (99.7%)**		**L210W**	**G190A**
				**T215Y (99.4%)**				**T215Y (99.8%)**			**T215Y**	
	**SE73**	Experienced	None	**D67N (44.2%)**	**K103N (95.9%)**	None	None	**D67N (49.2%)**	**K103N (96.1%)**	None	None	**K103N**
				**L210W (23.4%)**	**G190A (35.2%)**			**L210W (46.1%)**	**G190A (49.6%)**			**Y188C**
				**T215Y (19.7%)**				**T215Y (45.9%)**				
	**19C**	Naïve	**M46I (4.6%)**	None	None	None	Failed				None	None
	**20C**	Naïve	None	**T215S (2.1%)**	None	None	Failed				None	None
	**26C**	Naïve	None	**M184I (1.1%)**	None	None	Failed				None	None
	**05C**	Naïve	None	**K219R (2.3%)**	None	None	Failed				None	None
	**SE07**	Naïve	None	None	None	**Q148H (1.1%)**	None	None	None	None	None	None
	**SE14**	Naïve	None	None	None	**E138K (5.5%)**	None	None	None	**E138K (1.6%)**	None	None
	**SE31**	Naïve	**M46I (30.2%)**	None	None	None	**M46I (28.9%)**	None	None	None	None	None
	**SE97**	Naïve	**D30N (13.7%)**	None	None	None	**D30N (12.2%)**	None	None	None	None	None
	**24C**	Naïve	None	**T69D (99.7%)**	**M230L (99.7%)**	None	None	**D67E (77.2%)**	**M230L (99.8%)**	None	**T69D**	**M230L**
								**T69D (98.1%)**				
								**T215A (99.4%)**				
**Indian**	**IN04**	Naïve	None	None	**K101E (99.7%)**	None	None	None	**K101E (99.9%)**	None	NA	NA
	**IN05**	Naïve	None	**V75M (7.8%)**	None	None	None	**V75M (8.6%)**	None	None	NA	NA
	**IN06**	Naïve	None	None	**V106M (98.1%)**	None	None	None	**V106M (100.0%)**	None	NA	NA
					**Y181C (22.8%)**				**Y181C (23.6%)**			
	**IN07**	Naïve	None	**K65R (3.0%)**	**V106M (3.9%)**	None	None	**K65R (2.8%)**	**V106M (3.6%)**	None	NA	NA
				**K70E (3.4%)**	**G190S (95.6%)**			**K70E (3.2%)**	**G190S (96.1%)**			
				**L74V (66.1%)**				**L74V (67.6%)**				
				**M184V (7.5%)**				**M184V (8.4%)**				
				**M184I (3.7%)**				**M184I (3.2%)**				
				**T215I (3.4%)**				**T215I (3.3%)**				
	**IN08**	Naïve	None	None	**Y181C (99.5%)**	None	None	None	**Y181C (99.9%)**	None	NA	NA
**Ethiopian**	**ET160**	Naïve	**L76V (4.5%)**	None	None	None	**L76V (4.4%)**	None	None	None	None	None
	**ET115**	Naïve	None	**M184I (1.1%)**	None	None	None	None	None	None	None	None
	**ET122**	Naïve	None	None	**M230L (4.8%)**	None	None	None	None	None	None	None
	**ET155**	Naïve	None	None	None	**T66I (3.3%)**	None	None	None	**T66I (5.8%)**	None	None
	**ET159**	Naïve	None	**M184I (1.0%)**	None	None	None	None	None	None	None	None
	**ET171**	Naïve	None	**M184I (1.5%)**	None	**Q148H (1.2%)**	None	None	None	None	None	None

None → no DRM detected; Failed → failure due to upload limit of HTS file; NA → not available.

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
