# Peer review of "MiDRMpol: A High-Throughput Multiplexed Amplicon Sequencing Workflow to Quantify HIV-1 Drug Resistance Mutations against Protease, Reverse Transcriptase, and Integrase Inhibitors"

_viruses, 2019, doi:10.3390/v11090806_

Round 1
Reviewer 1 Report
The reviewers' comments on the original submission were addressed in this revision.
Reviewer 2 Report
Previous comments were addressed by the authors. The method description is now more detailed and script is publicly available.
This manuscript is a resubmission of an earlier submission. The following is a list of the peer review reports and author responses from that submission.
Round 1
Reviewer 1 Report
The manuscript describes a high-throughput sequencing analysis pipeline for detection of drug resistance mutations in HIV pol. The results that compare this pipeline with another, online PASeq, seem sound to me and show the advantage of the reported pipeline. I have only one major comment: the authors present this, even in the title and abstract, as a pipeline/workflow. They even claim that "it can be used without prior knowledge in HTS or bioinformatics". However, the description of the pipeline in methods is not detailed, and includes various previously published software tools (Cutadapt, FastUniq etc). Either the authors should make the complete script available (as text or available eg in GitHub), or they should include detailed description (parameters, diagram of workflow) in the manuscript.
Reviewer 2 Report
Manuscript ID: viruses-485287 Comments: The authors claimed that they have developed a new MiDRMpol pipeline that analyzes HIV HTS data for quantitative identification of DRMs residing in HIV pol gene. My comments are as following: Major comments: This paper reports a new MiDRMpol for HIV HTS data processing and quatitative DRM detection. As listed in Refs 3 and 30, many NGS-based HIV DR data analysis pipelines are already available. Although MiDRMpol could still be a good addition to the existing toolbox if it works well, the lack of novelty in this study is still a major concern. Reasons for this include: (1) Not much innovative data processing strategies were involved, or demonstrated, besides combining some existing tools for different analytical procedures; (2) Based on the presented information, I am not convinced about the accountability of outputs from this new tool. The gross error rates presented in Fig 1b is higher than those reprted in other NGS HIVDR studies using Illumina technologies (ie. MiSeq). (3) The comparability between MiDRMpol and PASeq is not consistent across varied DRM frequencies, and it is impossible to infer which one more accurately quantitative DRMs. The comparisons between GRT-PS and GRT-HTS have been performed in many previous studies. The relevant findings from this study are all expected. Although such data may still contribute, the manuscript should be focused more on the description and the performance assessment of the MiDRMpol tool. The authors claimed that this is a "simplified online pipeline for HTS data…" Therefore, the access information for this pipeline should be provided. Meanwhile, a summary of the exact quality control strategies applied in the pipeline should be presented and how they compare to those in PASeq may be discussed. Minor comments: The authors described a protocol that relies on PCR amplification of a fragment of >5000bp during HTS library prep. Notably, the longer the amplicon is, the lower success to amplify all templates present in HIV quasispecies, and eventually the less possibility to reliable detect minor variants. Did author assess this? At least, this should be mentioned in the discussion. Page 3, line 107: "… error rate…….was 0.18% (0.003-0.8%) (Fig 1B)" . This seems to be not consistent with what was shown in the figure. Figure 2: Is it based on the consensus from HTS reads? Why using >50% as cutoff? How were the 100 sequences selected? How much does it contribute to this paper reporting a new HTS analysis tool? Please clarify. Page 6, line 160: how were the 23 samples selected from all tested samples in the study? Although GRT-HTS detects HIV DRMs in a more sensitive way as compared to GRT-PS, generalized application of such methods for clinical DR monitoring is not feasible currently due to the high per sample cost. Such limitation should be acknowledged and the statement made in lines 227-229 on page 9 should be revised.
Title: MiDRMpol: A high-throughput multiplexed amplicon sequencing workflow to quantify HIV-1 drug resistance mutations against protease, reverse transcriptase and integrase inhibitors